# Genetic dissection of Nodal and Bmp signalling requirements during primordial germ cell development in mouse

Anna D. Senft[1], Elizabeth K. Bikoff[1], Elizabeth J. Robertson[1] & Ita Costello[1]

The essential roles played by Nodal and Bmp signalling during early mouse development have been extensively documented. Here we use conditional deletion strategies to investigate functional contributions made by Nodal, Bmp and Smad downstream effectors during primordial germ cell (PGC) development. We demonstrate that Nodal and its target gene Eomes provide early instructions during formation of the PGC lineage. We discover that Smad2 inactivation in the visceral endoderm results in increased numbers of PGCs due to an expansion of the PGC niche. Smad1 is required for specification, whereas in contrast Smad4 controls the maintenance and migration of PGCs. Additionally we find that beside Blimp1, down-regulated phospho-Smad159 levels also distinguishes PGCs from their somatic neighbours so that emerging PGCs become refractory to Bmp signalling that otherwise promotes mesodermal development in the posterior epiblast. Thus balanced Nodal/Bmp signalling cues regulate germ cell versus somatic cell fate decisions in the early posterior epiblast.

---

[1] Sir William Dunn School of Pathology, University of Oxford, Oxford OX1 3RE, UK. Correspondence and requests for materials should be addressed to E.J.R. (email: elizabeth.robertson@path.ox.ac.uk)

Primordial germ cells (PGCs), the precursors of sperm and eggs, are initially detectable in the early mouse embryo at around embryonic day (e) 6.25, prior to the onset of gastrulation[1]. Early fate mapping experiments revealed that the proximal posterior epiblast (PPE) gives rise to both the extra-embryonic mesoderm (ExM) and PGC cell populations[2]. The regulatory signals governing these cell fate decisions remain ill-defined. The PR domain containing zinc finger transcription factor Blimp1 (encoded by *Prdm1*) plays an essential role during PGC specification and expression at e6.25 identifies precursor PGCs (pre-PGCs)[3,4]. Commitment to the PGC lineage becomes evident slightly later between e6.75 and e7.5 when pre-PGCs activate expression of germ cell markers such as Stella (*Dppa3*) and Ap2γ (*Tfap2c*), concomitantly reactivate expression of pluripotency genes, including Sox2 and repress somatic gene expression[1,5]. By e8.5, specified PGCs have migrated from the base of the allantois into the overlying endoderm, and subsequently migrate along the dorsal hindgut endoderm before homing to and colonising the genital ridges from e10.5[1]. Extensive chromatin remodelling and genome-wide epigenetic reprogramming occurs during migration[6]. Transcriptional profiling experiments analysing PGCs in vivo and in vitro differentiated PGC-like cells (PGCLCs) have provided insights into the dynamic transcriptional changes that accompany PGC maturation[5,7,8].

Correct patterning of the early post-implantation stage embryo depends on reciprocal signalling cues by members of the TGFβ family of secreted growth factors controlling cell–cell interactions between the pluripotent epiblast and the overlying extra-embryonic tissues namely the extra-embryonic ectoderm (ExE) and the visceral endoderm (VE)[9,10]. Temporally and spatially restricted expression of Nodal and Bmp ligands results in activation of their cognate receptors, phosphorylation of the intracellular effectors, Smad2 and Smad3 (Smad23), or Smad1, Smad5 and Smad9 (Smad159), respectively, that associate with the co-Smad Smad4, translocate to the nucleus and activate cell-type specific target gene expression[11]. Nodal signalling in the early epiblast is required for correct patterning of the VE, formation of the anterior–posterior (A–P) axis and also promotes development of the ExE[12]. Together with Bmp signals from the ExE, Nodal initiates mesoderm induction and primitive streak (PS) formation within the PPE[12,13]. During gastrulation, graded Nodal signals pattern the PS[10]. Thus, highest levels of Nodal signalling are required for specification of anterior PS derivatives. Lowering Nodal expression levels in the PS, or depleting Smad23 in the epiblast results in failure to correctly specify the anterior definitive endoderm (DE) and the embryonic midline[14].

In contrast Bmp/Smad159 signals promote the formation of ExM. Loss of Bmp4 from the ExE disrupts gastrulation and is associated with truncation of posterior structures, including the allantois and yolk sac[13]. Both Smad1 and Smad5 null embryos also display ExM tissue defects[15,16]. Mutant embryos lacking Bmp4 or Bmp8b expression in the ExE or those lacking Bmp2 in the VE display compromised PGC development in the epiblast[17–19]. Bmp4 null embryos entirely lack mature PGCs, while Bmp4 heterozygous embryos contain reduced numbers of PGCs. The observation of reduced numbers of PGCs in embryos lacking either Smad1 or Smad5[15,16], as well as in Smad1/5 double heterozygous embryos[20], provides convincing evidence that dose-dependent Bmp signalling governs PGC specification. The Wnt signalling pathway also regulates PGC development[21]. Wnt3 is normally induced in the posterior epiblast in response to Nodal/Bmp signalling[22]. Wnt3 mutants fail to gastrulate[23] and lack a detectable pre-PGC population[21]. Collectively, these findings demonstrate that patterning during gastrulation and PGC formation are co-ordinately regulated by dynamic signalling events. However, given the complex morphological disturbances

observed in loss of function mutant embryos, it has proven difficult to further dissect the crosstalk between the embryonic and extra-embryonic tissues. In particular since Nodal null embryos arrest prior to gastrulation[24], any possible role of Nodal signalling during PGC specification remains to be explored.

Here, we exploit tissue specific conditional deletion strategies to investigate functional contributions made by Nodal and Bmp signalling within the embryonic versus extra-embryonic tissues. We directly assess the distinct roles played by various signalling pathway components in the VE and the epiblast during formation of the PGC lineage. Collectively, our experiments provide insights into the signalling cues that cooperatively regulate the size of the founding pre-PGC population and govern the PGC developmental programme during induction, specification and migration at early post-implantation stages of mouse development.

## Results

**Smad2 in the VE restricts Bmp signals to the proximal region.** In Smad2 null embryos, lacking the anterior VE (AVE) signalling centre, the epiblast adopts an exclusively ExM fate[12,25]. These mis-patterned embryos arrest at around e9.5. Here, to examine cell-type specific Smad2 functional contributions within the VE, we crossed animals carrying a conditional *Smad2* allele (*Smad2CA*) with heterozygous *Smad2*[+/−] mice carrying the Ttr-Cre transgene[26] (Supplementary Figure 1A). As shown below, we found that the Smad2ΔVE embryos phenocopy the *Smad2*[−/−] embryos, strengthening the idea that the dramatic tissue disturbances observed in the null embryos predominantly reflect the loss of Smad2 signalling in the AVE.

To assess the possible impact on Bmp signalling, we analysed phospho-Smad159 (p-Smad159) localisation. At e5.5 p-Smad159 expression is normally restricted to the proximal VE. However, p-Smad159 staining in Smad2ΔVE e5.5 embryos is detectable throughout the entire VE, including the distal region (Fig. 1a). Whole-mount in situ hybridisation experiments demonstrate that *Bmp4* expression remains unchanged, whereas Smad2ΔVE embryos lack *Bmp2* transcripts (Supplementary Figure 1B). Thus, higher levels of p-Smad159 cannot simply be explained due to increased expression of Bmp ligands.

Antagonistic Bmp and Nodal signalling cues govern VE specification[9]. However, the regulatory mechanisms that normally restrict p-Smad159 signalling to the proximal VE have yet to be fully characterised. The TGFβ antagonist Gdf3, expressed in the epiblast and distal VE, directly antagonises Bmp4 activity[27,28]. Moreover, selective mesoderm expansion in double homozygous embryos lacking both *Gdf3* and the closely related ligand *Gdf1* has been documented[29]. Here, we observe in Smad2ΔVE embryos that *Gdf3* expression is absent in the VE and reduced in the epiblast (Fig. 1b). Thus, up-regulated p-Smad159 activity in Smad2ΔVE embryos potentially reflects decreased *Gdf3* expression levels.

**Loss of *Smad2* from the VE expands the PGC niche.** Alkaline phosphatase (AP) positive presumptive PGC clusters were previously identified in e8.5 Smad2[−/−] embryos[15]. PGCs are formed within the PPE and are reliant on Bmp signalling from the adjacent ExE for their development. To evaluate if and when PGCs are formed in Smad2ΔVE embryos, where there is an excess of Bmp signalling, we examined PGC marker gene expression. Nanog, normally reactivated in the early proximal epiblast[30], is also strongly expressed in developing PGCs[31]. As expected in control embryos, we detected cells co-expressing Nanog and the pluripotency marker Oct4 in the PPE (Fig. 1c). Similarly, at e6.5 Smad2ΔVE embryos contain Nanog/Oct4

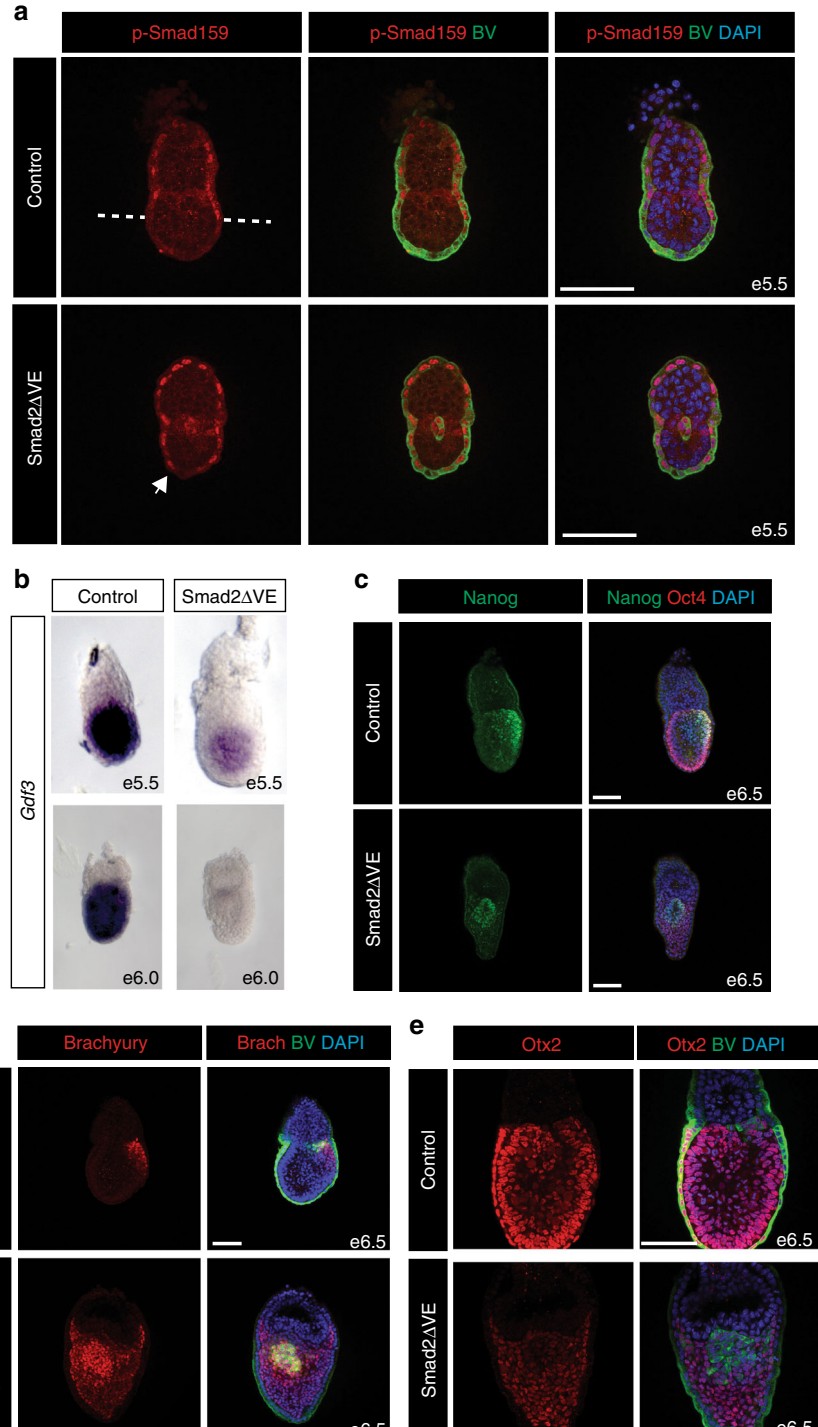

**Fig. 1** Imbalanced Bmp/Nodal signalling and expansion of the PGC niche caused by Smad2 inactivation in the VE. **a** Representative images of p-Smad159 immunofluorescence (IF) staining of e5.5 embryos carrying the Blimp1-mVenus (BV) transgene, counter stained with DAPI. Dashed line indicates extent of proximal p-Smad159 staining in control embryos. The arrow indicates expanded p-Smad159 in the distal VE of Smad2ΔV embryos. **b** Whole-mount in situ hybridisation analysis of *Gdf3* expression in control and Smad2ΔVE embryos at e5.5 and e6.0. **c** Nanog and Oct4 co-staining in e6.5 control and Smad2ΔVE embryos. **d** Brachyury IF in e6.5 control and Smad2ΔVE BV-expressing embryos. **e** Otx2 staining and BV expression at e6.5. All IF staining images were counter stained with DAPI. Scale bars = 100 μm

double positive cells adjacent to the ExE, but these were located in a central position in the epiblast (Fig. 1c).

Next, to assess whether these cells correspond to pre-PGCs, we used the membrane tethered Blimp1-mVenus (BV) BAC transgene that faithfully recapitulates Blimp1 expression in both the VE and the developing PGCs[32]. Smad2ΔVE mutants expressing the BV transgene clearly contain BV-positive (BV+) pre-PGCs that also co-express E-Cadherin (Supplementary Figure 1C). As judged by immunohistochemistry these cells strongly express endogenous Blimp1 protein (Supplementary Figure 1D). Interestingly, in Smad2ΔVE embryos BV+ pre-PGCs are initially detectable within the epiblast at e5.5, 12–18 h before

their appearance in wild-type embryos (Fig. 1a). Slightly later at e6.5, the Smad2ΔVE proximal epiblast contains increased numbers of BV[+] cells as compared to control wild-type embryos (Fig. 1d, e).

BV[+] cells in the proximal epiblast at e6.5 normally express Brachyury and retain E-Cadherin expression. In contrast the adjacent mesodermal cells also strongly express Brachyury but down-regulate E-Cadherin expression (Fig. 1d, Supplementary Figure 1C). The central core of BV[+] epithelial cells in Smad2ΔVE embryos is likewise surrounded by Brachyury positive-mesodermal cells (Fig. 1d). At e6.5, in both control and Smad2ΔVE embryos BV[+] cells weakly express Otx2, Eomes and Sox2 (Fig. 1e, Supplementary Figure 1E and F). Slightly later at e7.5 BV[+] cell clusters retain E-Cadherin and co-express Stella as well as Oct4 (Supplementary Figure 1G and H). Overall, in Smad2ΔVE embryos we observe increased numbers of BV, Oct4, Nanog and Stella co-expressing PGCs surrounded by Brachyury-positive, E-Cadherin-negative mesodermal cells. Thus, Smad2 expression in the VE normally restricts expansion of the PGC niche so that only a small subset of posterior epiblast cells become allocated to a PGC fate.

Next, to examine Smad2 functional contributions within the epiblast we made use of the Sox2-Cre deleter strain[33]. Stella[+] PGCs are readily detectable at e8.5 in Smad2ΔEpi embryos, suggesting that Smad2 is dispensable for PGC specification (Supplementary Figure 2A). However, it seems likely that the closely related Smad3 effector, known to be robustly expressed in the epiblast[14], functionally compensates.

Blimp1 is strongly expressed in the VE, but experiments to date have failed to demonstrate any VE-specific functional contributions. To test whether Blimp1 VE expression may influence PGC specification, we used the Ttr-Cre deleter strain. As shown in Supplementary Figure 2C, selective loss of Blimp1 in the VE has no noticeable effect on the formation of Blimp1-expressing PGCs at the base of the allantois (Supplementary Figure 2B and C). Moreover, Prdm1ΔVE mice are viable and fertile (Supplementary Table 1).

**Both Nodal and Eomes are essential during PGC specification**. Smad2 and the closely related Smad3 are intracellular effectors of the Nodal pathway. Nodal null ES cells, as well as Smad23 double mutant ES cells, fail to generate PGCLCs in vitro[34,35]. To investigate Nodal functional contributions in vivo we examined formation of pre-PGCs in Nodal[−/−] embryos. At e5.5 Nanog expression levels are dramatically reduced as compared to control embryos (Fig. 2a, Supplementary Figure 2D). However, Nodal[−/−] embryos robustly express the epiblast marker Oct4 (Supplementary Figure 2D). Interestingly, Nodal[−/−] embryos display precocious induction of BV[+] epiblast cells (Fig. 2a). These BV[+] cells have either weak or no Nanog expression at e5.5 (Fig. 2a). By e6.5 there is no expression of Nanog in the BV[+] epiblast cells or indeed any other cells of the epiblast (Supplementary figure 2E). At e6.5 the expanded BV[+] cell population in Nodal[−/−] embryos lack precocious expression of specified PGC markers, such as Ap2γ (Supplementary Figure 2F), even though the BV transgene was prematurely expressed. Notably at e6.5 Nodal mutants do not maintain *Bmp4* expression[12], hence an important germ cell inducing signal is also absent in these embryos. Accordingly, at e7.5 Ap2γ/ BV[+] co-expressing cells are not observed in Nodal[−/−] embryos (Supplementary Figure 2G). Hence, specified PGCs are not observed in Nodal[−/−] embryos. As for Smad2ΔVE embryos, e5.5 Nodal mutants similarly exhibit increased levels of p-Smad159 staining in the distal VE (Fig. 2b). Moreover p-Smad159 is also detectable throughout the epiblast (Fig. 2b). These results suggest that Nodal functions in vivo to promote optimal levels of

Nanog during pre-PGC development and also to spatially restrict the PGC niche.

The T-box transcription factor Eomes, acting downstream of Nodal signalling, plays essential roles during gastrulation[36,37]. Eomes is also required to pattern the VE[38]. As shown in Supplementary Figure 2H, EomesΔVE embryos contain Blimp1[+] epiblast cells. Thus, Eomes activity in the VE is non-essential for Blimp1 induction in the epiblast. Conditional deletion in the epiblast (EomesΔEpi) results in defective epithelial-to-mesenchymal transition (EMT) and the failure of nascent mesoderm cells to down-regulate E-Cadherin and exit the PS[36]. Eomes is expressed in early pre-PGCs, but becomes down-regulated by late streak stages (Supplementary Figure 1E, Supplementary Figure 2I). Interestingly, EomesΔEpi e7.5 embryos contain an expanded epithelial-like BV[+] cell population (Fig. 2c) that also express endogenous Blimp1 protein (Supplementary Figure 2H). Nanog expression is reactivated within the epiblast, however, the BV[+] cell population in EomesΔEpi embryos is largely Nanog negative (Fig. 2d). Moreover, at e7.5 BV[+] cells fail to activate the PGC markers Ap2γ, Sox2 and Stella (Fig. 2e) and only a subset retain Brachyury activity (Fig. 2f). These results demonstrate that the loss of the T-box transcription factor Eomes in the epiblast disrupts the PGC developmental programme in vivo.

**Smad1/4 roles in PGC specification and maintenance or migration**. Smad1 is an intracellular effector of the Bmp signalling pathway, while the co-Smad, Smad4, is important for transducing aspects of both Nodal and Bmp signalling. Smad1 and Smad4 have previously been implicated in PGC development, as both Smad1 null and Smad4ΔEpi embryos lack AP-positive migrating PGCs at e8.5[15,39]. However, cooperative or possibly unique functional roles at distinct developmental stages during PGC lineage specification have yet to be examined. Initially to explore Smad4 functional requirements in the VE, we generated Smad4ΔVE embryos. We found at e7.5 that Blimp1-expressing PGCs are formed appropriately at the base of the allantois (Supplementary figure 3A). Similarly, Smad4 activity in the epiblast is non-essential for PGC specification. BV/Stella co-expressing cells are detectable on the posterior side of Smad4-ΔEpi embryos at e7.5 (Supplementary figure 3B). However, at e8.5 these Stella/ BV[+] PGCs remain at the base of the allantois and fail to migrate towards the gut endoderm (Fig. 3a). In addition, we used a Blimp1-Cre deleter strain[40] to selectively eliminate Smad4 expression within the pre-PGC cell population (Smad4ΔPGC). Relatively, few Stella expressing cells were present in e8.5 Smad4ΔPGC embryos (Supplementary Figure 3C). These results demonstrate that Smad4 is dispensable for initial PGC specification, but is required for PGC maintenance and/or migration.

Smad1[−/−] embryos form only a rudimentary allantoic bud and develop a distinctive out-pocketing of the proximal posterior VE[15]. At e7.5, only a few BV/Oct4[+] pre-PGCs are detectable underneath the ruffled VE (Fig. 3b) and mature Stella positive PGCs fail to emerge[15]. To further investigate Smad1 functional requirements in the VE, we used the Ttr-Cre deleter strain. Smad1ΔVE embryos display extensive ruffling of the VE (Supplementary Figure 3D). However, Blimp1[+] pre-PGCs are occasionally observed in the early epiblast (Supplementary Figure 3E). An epiblast specific Smad1 deletion likewise results in VE ruffling and the appearance of rare BV/Oct4 double positive cells underneath the overgrown VE (Fig. 3c). However, as for *Smad1* null embryos these BV/Oct4[+] cells fail to form specified PGCs. The ability to generate some BV/Oct4[+] cells may be due to Smad5 compensation, which is also expressed in the

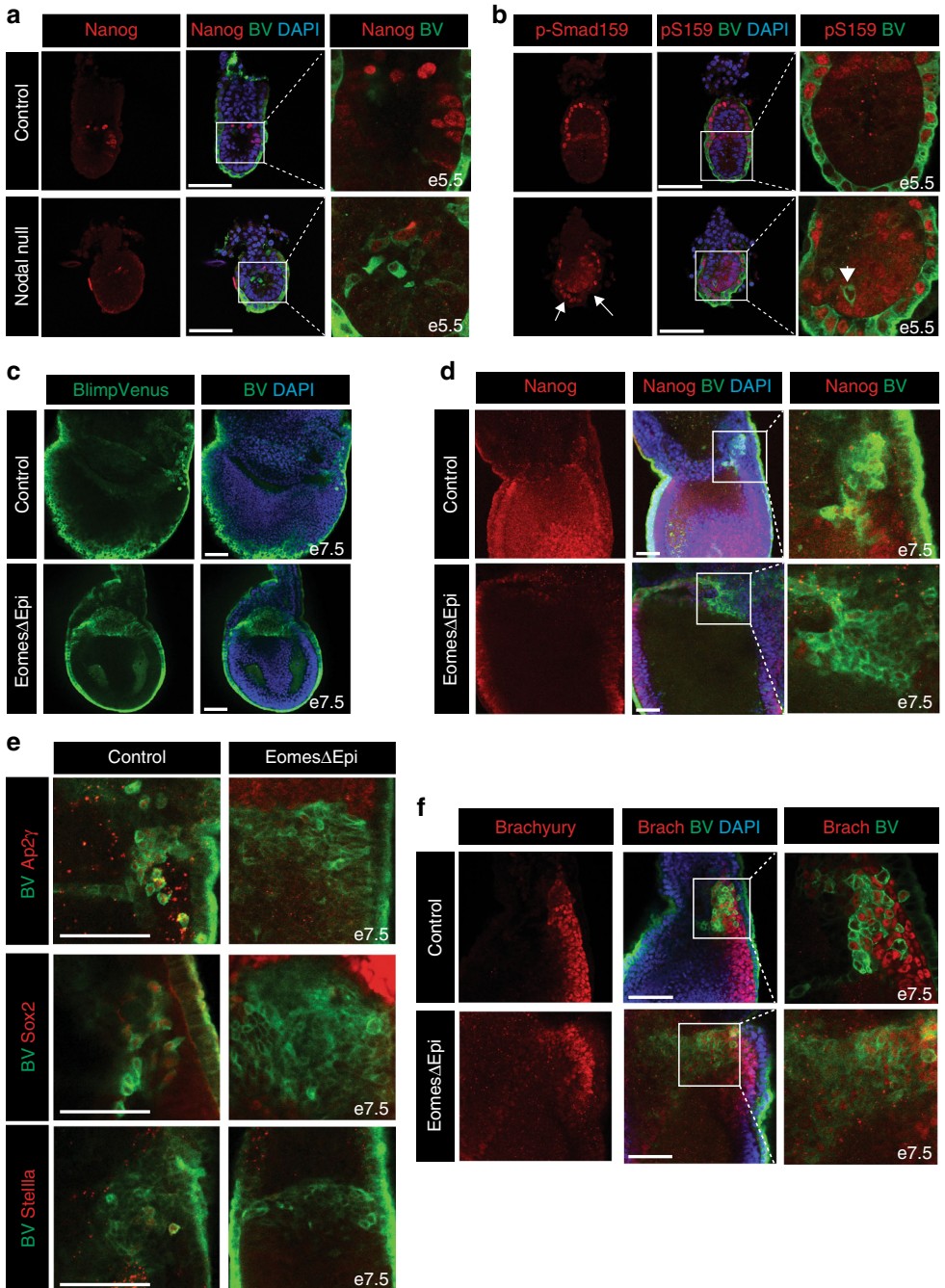

**Fig. 2** Nodal and its downstream target Eomes regulate early stages of PGC development. **a** Reduced levels of Nanog expression in e5.5 Nodal null BV+ embryos. **b** Nodal null BV+ embryos display an expansion of p-Smad159 staining in the distal VE, as indicated by arrows. Arrowhead indicates lowered p-Smad159 staining within BV+ cells. **c** Expansion of the BV+ cell population in EomesΔEpi embryos at e7.5. **d** Nuclear Nanog staining in EomesΔEpi and control BV+ embryos at e7.5. **e** Analysis of Ap2γ, Sox2 and Stella in e7.5 control and EomesΔEpi BV+ cells. **f** Brachyury staining in control and EomesΔEpi BV+ e7.5 embryos. All IF images are counter stained with DAPI. Scale bars = 100 μm

epiblast of gastrulating embryos[15,20]. Indeed, in addition to reduced PGC numbers in embryos lacking either Smad1 or Smad5, Smad1/5 double heterozygotes also have a reduced number of PGCs[20], revealing a critical role for both Smad1 and Smad5 in germ cell development.

Taken together these experiments demonstrate that Smad1 functions in the epiblast during PGC specification, whereas in contrast Smad4 promotes PGC maintenance and/or migration. Hence, the Smad4ΔEpi mutant phenotype is intermediate to the Smad2ΔEpi and Smad1−/− phenotypes, as in Smad2ΔEpi embryos Stella+ cells form at the posterior side, while Smad1

mutants initiate pre-PGC differentiation but never develop specified PGCs.

**Nodal and Bmp signalling requirements during PGCLC formation.** Recent reports have described embryonic stem (ES) cell culture protocols for in vitro differentiation of PGCLCs that, as judged by gene-expression patterns and global epigenetic remodelling profiles, closely resemble bona fide PGCs[7,8]. To further investigate Nodal and Bmp signalling requirements we generated Smad2−/−, Eomes−/−, Smad1−/−, Smad4−/− and Wnt3−/− ES

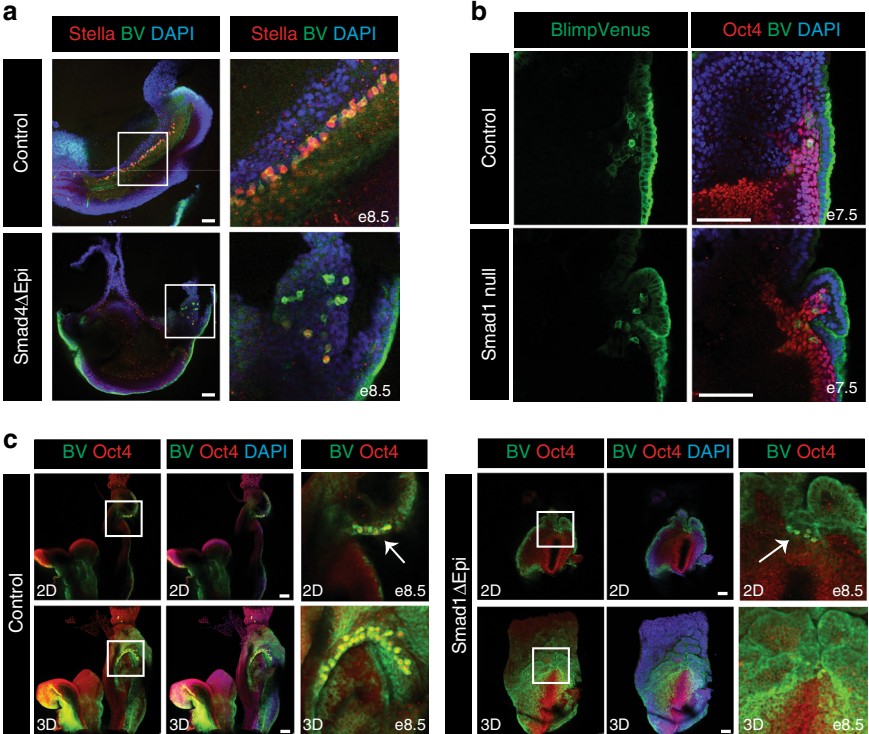

**Fig. 3** Smad1 is required for PGC specification whereas Smad4 controls PGC maintenance and migration. **a** Stella staining at e8.5 shows PGCs migrating along the hindgut endoderm in BV⁺ control but not Smad4ΔEpi embryos. **b** Oct4 staining in Smad1 null and control e7.5 BV⁺ embryos. **c** Single optical sections (2D) and z-stack projections (3D) showing Oct4 staining in the posterior side of Smad1ΔEpi and control BV⁺ embryos. Arrows indicate Oct4 and BV co-expressing cells. All IF images are counter stained with DAPI. Scale bars = 100 μm

cell lines carrying the BV transgene. Control and mutant BV⁺ ES cells, grown under naïve conditions, were induced to form epiblast-like cells (EpiLCs), and subsequently aggregated under appropriate conditions to yield PGCLCs (Fig. 4a). As expected in wild-type cultures at day 2 and day 4 of PGCLC differentiation BV/ Stella, BV/Oct4, as well as BV/Ap2γ co-expressing cells were detectable by immunofluorescence (Fig. 4b, Supplementary Figure 4A and B). As a negative control, consistent with previous studies[41], Wnt3⁻/⁻ cultures contained only a few double positive PGCLCs (Fig. 4b, Supplementary Figure 4A and B). Our Smad2⁻/⁻, Eomes⁻/⁻, Smad1⁻/⁻ and Smad4⁻/⁻ ES cell lines yielded BV/Stella, BV/Oct4 and BV/Ap2γ double positive PGCLCs (Fig. 4b, Supplementary Figure 4A and B). However, as judged by immunofluorescence and real-time quantitative polymerase chain reaction (RT-qPCR) analysis, the efficiency of PGCLC induction was compromised as compared to wild-type control cultures. Thus, Smad2⁻/⁻, Eomes⁻/⁻ and Smad4⁻/⁻ cell aggregates express lower levels of *Prdm1* (Blimp1) and *Tfap2c* (Ap2γ), while Smad1⁻/⁻ cultures show reduced *Tfap2c* and increased levels of the somatic marker *Hoxb1* (Fig. 4c). These results demonstrate that in contrast to the situation in vivo, in cultures containing high levels of exogenously added growth factors, Eomes⁻/⁻ and Smad1⁻/⁻ ES cells can differentiate into PGCLCs.

**Decreased Bmp responsiveness accompanies PGC specification.** At e6.5 p-Smad159 staining is normally restricted to the extra-embryonic VE, the anterior VE and the PPE whereas the posterior embryonic VE lacks p-Smad159 (Fig. 5a). However, in Smad2ΔVE embryos, p-Smad159 is detectable throughout the entire VE at e5.5 (Fig. 1a), but slightly later at e6.5 the VE does not have any p-Smad159 reactivity (Fig. 5a). The mesodermal cell population in e6.5 Smad2ΔVE embryos surrounding BV⁺ cells

display strong p-Smad159 immunoreactivity. However, the BV⁺ cells themselves have substantially reduced levels of p-Smad159 staining (Fig. 5a). Likewise, the BV high-expressing cells in control e7.5 embryos, as well as the expanded numbers of BV⁺ cells present in e7.5 Smad2ΔVE embryos, lack p-Smad159 reactivity (Fig. 5b). These results demonstrate that down-regulated p-Smad159 levels distinguishes specified PGCs from their somatic neighbours and that specified PGCs do not respond to high levels of Bmp signalling. Additionally, EomesΔEpi embryos also display a high proportion of BV⁺ epiblast cells lacking p-Smad159 activity (Supplementary Figure 5A). Decreased p-Smad159 staining was also observed within the BV⁺ cells generated during in vitro PGCLC differentiation (Fig. 5c, Supplementary Figure 5B). Thus, decreased responsiveness to Bmp signalling is a characteristic feature of PGC cell populations, in both in vitro and in vivo settings.

To directly assess whether Blimp1 expression influences Smad159 phosphorylation levels, we generated Blimp1⁻/⁻ embryos carrying the BV transgene. BV⁺ epiblast cells were visible at the base of the allantoic bud at e7.5 in Blimp1⁻/⁻ embryos (Fig. 5d), strengthening the notion that Blimp1 functional activity is not required at early stages during pre-PGC formation. Interestingly a subset of BV⁺ cells display decreased p-Smad159 levels, while others retain p-Smad159 reactivity (Fig. 5d). To test whether loss of p-Smad159 within the maturing PGCs potentially reflects reduced levels of transcription we examined RNA-seq data sets[42,43] (Supplementary Table 3). Reduced *Smad1* and *Smad5* expression levels were reported in e7.5 PGCs as compared to the surrounding mesodermal cell population (Supplementary figure 5C). Moreover RNA-seq analysis of wild-type PGCs[43] confirms that *Smad1* expression is down-regulated during PGC development (Supplementary Figure 5D), whereas in contrast BV⁺ Blimp1⁻/⁻ PGCs fail to down-

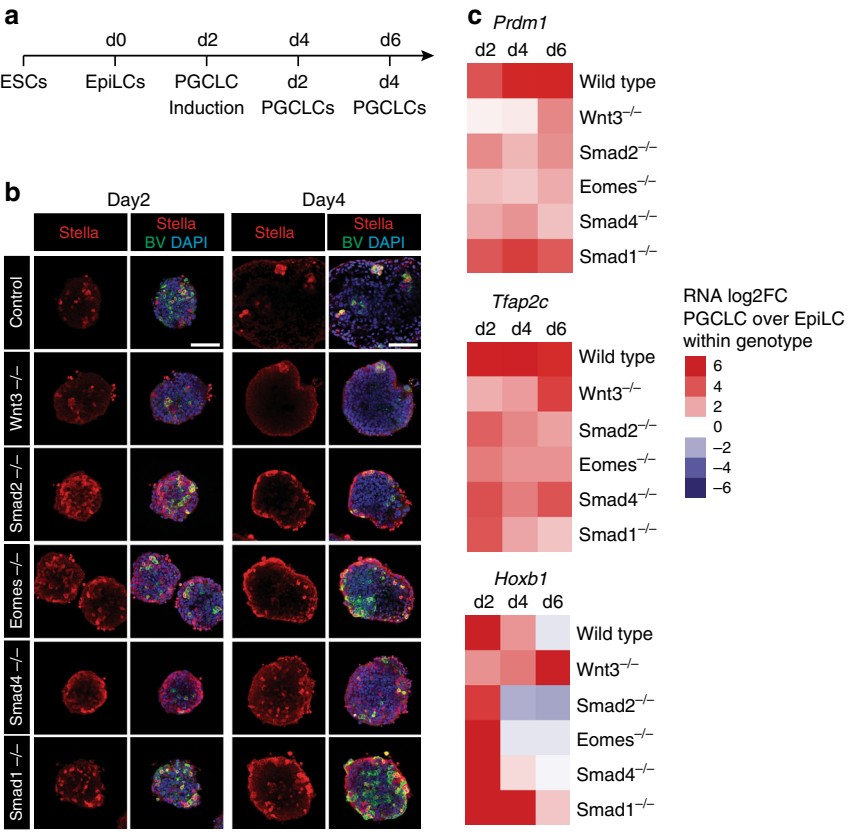

**Fig. 4** Abilities of mutant ES cells lacking Nodal/Bmp/Smad pathway components to undergo PGCLC differentiation. **a** Time-line and culture protocol used for PGCLC differentiation. **b** Stella IF staining of day 2 and day 4 BV[+] PGCLC aggregates derived from the indicated genotypes, counter stained with DAPI. Scale bars = 100 μm. **c** Heatmap showing the log2 fold changes (log2FC) of *Prdm1, Tfap2c* and *Hoxb1* expression, as judged by RT-qPCR, at day 2, 4 and 6 of PGCLC differentiation in control and mutant cells of the indicated genotypes, relative to EpiLCs

regulate *Smad1* expression (Supplementary Figure 5D). Consistent with this recent Blimp1 ChIP-Seq experiments analysing d6 PGCLCs[43] (Supplementary Table 3) allowed us to identify a Blimp1 peak downstream of the *Smad1* first coding exon containing the canonical Blimp1 binding motif (Fig. 5e). Collectively, these observations suggest that Blimp1 may directly repress *Smad1* transcription during early PGC development. Conditional deletion of Blimp1 in mature PGCs, using the Stella-Cre deleter strain[43], has no effect on *Smad1* transcriptional levels (Supplementary Figure 5D). Thus, Blimp1 may down-regulate *Smad1* expression at early stages of PGC development, potentially ensuring that the emerging PGCs are refractory to the high-Bmp signalling environment in the posterior epiblast that normally promotes development of the ExM.

## Discussion

The transcriptional regulators and epigenetic machinery responsible for guiding allocation of the PGC lineage have been intensively investigated over the past decade[1,5]. The small population of pre-PGCs (~5) initially present at e6.25 in the proximal epiblast of pre-streak embryos are marked by expression of the zinc finger transcriptional repressor Blimp1[3,44], together with the pluripotency factors Oct4 and Nanog. Slightly later at e6.75, *Prdm14* expression is induced in developing PGCs[45] and subsequently, coincident with formation of the posterior ExM, PGCs activate expression of *Dppa3* (Stella), *Tfap2c* (Ap2γ) and *Sox2*. The cluster of specified PGCs (~40) is detectable at e7.5 within the ExM at the base of the allantois. Over the following 24 h these committed PGCs migrate into the endodermal layer of the forming hindgut pocket.

Dose-dependent Nodal/Smad functional activities are known to be required for A–P and left–right axis patterning in the early mouse embryo[10]. Functional contributions during gastrulation and within the PS derivatives have been extensively characterised[10]. Here, we dissected Nodal/Bmp/Smad signalling cues within embryonic and extra-embryonic tissues that govern the PGC developmental programme. The present experiments now reveal importantly that Nodal/Smad2 activities also regulate allocation of the pre-PGC population. Thus Nodal null embryos prematurely initiate pre-PGC formation at e5.5. However, these BV[+] epiblast cells are ectopically positioned and lack Nanog and Ap2γ expression. Nanog, an important regulator of PGC development in vivo[46,47], has been previously been shown to induce PGCLC differentiation in vitro[48]. Nanog directly upregulates both *Prdm1* (Blimp1) and *Prdm14* expression in PGCLCs[48]. However, over-expression activates *Prdm14* expression prior to *Prdm1*[48] in EpiLCs, whereas *Prdm1* expression in pre-PGCs in vivo precedes that of *Prdm14*[45,48]. Thus Nanog re-expression downstream of Nodal seems to be non-essential for induction of *Prdm1* expression per se. Rather Nanog may primarily function to activate *Prdm14*. The present results demonstrate that Nodal signalling in the proximal epiblast, prior to the onset of streak formation, is required to fully reactivate and maintain high levels of Nanog expression.

Nodal is transiently expressed in the proximal epiblast and expression decreases as the streak elongates[49]. The high levels retained in the anterior streak function to induce DE and midline fates[10]. Mutant embryos lacking expression of its downstream effector Smad2 fail to induce the antagonists *Cer1* and *Lefty1* in the VE and consequently display ectopic Nodal signalling

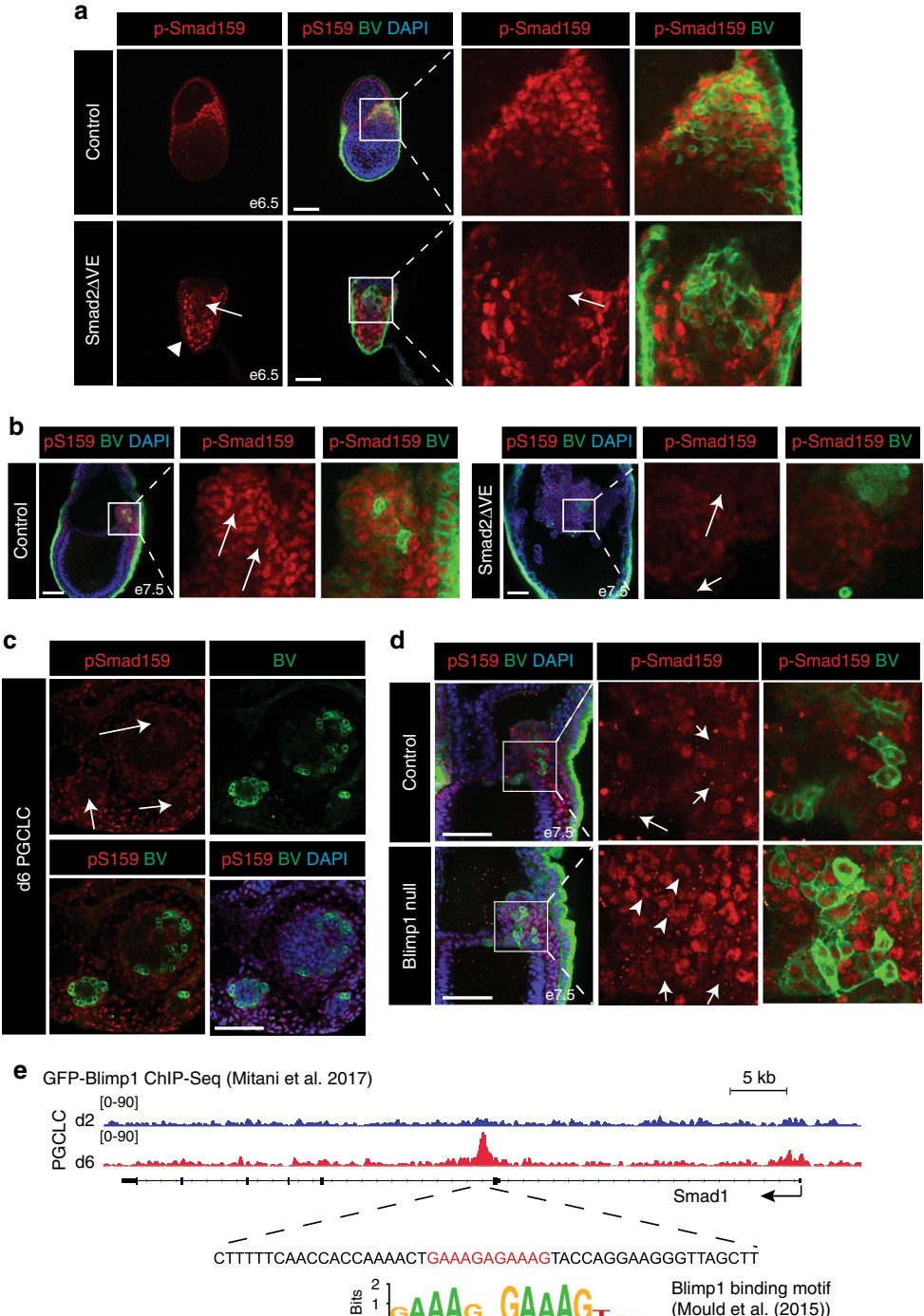

**Fig. 5** PGCs down-regulate p-Smad159 to become refractory to the high-Bmp signalling environment. **a** p-Smad159 staining of BV+ cell populations. Arrowhead confirms e6.5 Smad2ΔVE embryos lack p-Smad159 in the VE, while arrows indicates weak activity in BV-expressing epiblast cells. **b** p-Smad159 staining in control and Smad2ΔVE BV+ embryos at e7.5. Arrows indicate loss of p-Smad159 staining in BV high-expressing cells. **c** p-Smad159 staining in BV+ wild-type d6 PGCLC aggregates. Arrows indicate BV+ cell populations lacking p-Smad159 reactivity. **d** p-Smad159 staining in control and Blimp1−/− BV+ embryos at e7.5. Arrows indicate reduction of p-Smad159 reactivity in BV high-expressing cells in expanded panels. Arrowheads indicate retained p-Smad159 activity in BV+ cells in the Blimp1−/− mutant embryo. All IF images are counter stained with DAPI. Scale bars = 100 μm. **e** Genome browser track view of GFP-ChIP wiggle plots from GFP-Blimp1 expressing PGCLC cultures at day (d)2 (blue) and d6 (red) showing enrichment of GFP-Blimp1 density at a *Smad1* intronic region. The sequence below the GFP-Blimp1 ChIP site contains a Blimp1 binding motif indicated in red. The consensus Blimp1 binding motif previously identified by genome-wide ChIP-seq data sets is also indicated[65]

throughout the entire epiblast[12,25]. The present experiments demonstrate that conditional Smad2ΔVE mutants phenocopy Smad2 null embryos. As judged by down-regulated expression of Otx2 and Sox2, together with up-regulated Brachyury expression, the mutant epiblasts appear to prematurely adopt a proximal posterior character. An inverse relationship between p-Smad159 and p-Smad23 signalling in the VE has previously been characterised[9]. Here, we found that e5.5 Smad2ΔVE embryos display ectopic p-Smad159 activation throughout the VE. Interestingly, these disturbances in Smad2ΔVE mutants are associated with

decreased expression of the Bmp antagonist *Gdf3* and likely accounts for the increased levels of p-Smad159 in the VE. Thus imbalanced Nodal/Bmp signalling leads to the precocious appearance and markedly increased numbers of pre-PGCs at e6.5. These results demonstrate Nodal/Smad2 signals localised to the proximal epiblast govern competency to acquire PGC characteristics and promote the establishment of the PGC niche.

Recent work suggests that occupancy at a putative T-box site, 10 kb upstream of the *Prdm1* transcription start site, by the T-box transcription factor family member Brachyury activates Blimp1 expression in the pre-PGC population[41]. Thus Brachyury mutant embryos activate but fail to maintain BV⁺ pre-PGCs[41]. Consistent with this, forced expression of either Brachyury or Eomes in EpiLCs leads to up-regulated *Prdm1* expression[41], suggesting that Eomes and Brachyury may function redundantly to activate *Prdm1* expression. However, the present experiments demonstrate that Eomes is dispensable for Blimp1 induction in the epiblast. Moreover, the BV⁺ cells formed in the absence of Eomes mostly lack Brachyury expression, strongly suggesting that Brachyury and Eomes are dispensable for initial induction of Blimp1.

Eomes in the early PS directly activates *Mesp1* expression necessary to allow nascent mesoderm to down-regulate E-Cadherin and undergo EMT[36,50]. Here, we demonstrate that Eomes, acting immediately downstream of Nodal within the epiblast[37], plays an essential role during the early stages of PGC formation. In the absence of Eomes, we observe an expanded population of BV⁺ cells in the proximal epiblast. However, these pre-PGCLCs lack Nanog expression and fail to activate the germ cell programme. Interestingly, Eomes mutant BV⁺ cells, in close proximity to Bmp4 signals from the overlying ExE, lack p-Smad159 activity and are refractory to Bmp signalling. The absence of Eomes in the proximal epiblast disturbs formation of the PGC niche within the posterior ExM population that normally promotes PGC specification and survival[51,52]. Previous work shows that Eomes null PS cells can undergo EMT and generate mesodermal cell populations in vitro[36]. Here, we demonstrate that Eomes null ES cells can differentiate into BV/Oct4/Ap2γ/Stella positive PGCLCs, albeit at a lower efficiency as compared to wild-type controls, most likely due to their ability to generate a supporting mesodermal population. Thus, Eomes is not intrinsically required for Blimp1 induction and PGC formation per se. Rather, acting downstream of Nodal, Eomes is essential to promote formation of the posterior signalling niche necessary for maturation of the PGC lineage.

How PGCs maintain their unique cell-type identity within the predominant mesodermal signalling environment at the base of the allantois remains mysterious. Here, we describe an unappreciated cellular mechanism that allows PGCs to insulate themselves and become refractory to Bmp signals. Thus, we found at e7.5 that BV⁺ cells display reduced Smad159 phosphorylation levels. Moreover within PGCLC aggregates p-Smad159 activity and BV expression are mutually exclusive. Down-regulated expression of both *Smad1* and *Smad5* was previously observed in PGCs compared to the surrounding mesoderm[42]. Blimp1 may directly repress *Smad1* transcription via occupancy at a site within the Smad1 locus. Consistent with this idea recent RNA-seq data sets[43] reveal that *Smad1* expression in Blimp1-deficient pre-PGCs is not down-regulated.

The cluster of PGCs at the base of the allantois, that themselves lack Bmp4 expression, are surrounded by ExM expressing high levels of Bmp4[17]. While PGCs themselves become refractory to the Bmp signalling environment, their survival and further development is thought to be critically dependent on the ability of the adjacent ExM to provide extrinsic trophic growth factors[52]. Recent in vitro experiments similarly suggest that the role of Smad1 signalling during PGCLC differentiation may be to

generate somatic cells that function to promote PGCLC survival[53]. Likewise, we observe here in PGCLC aggregates localised pockets of BV⁺ cells within p-Smad159 active regions. Thus, a continuum of Bmp signalling in the early post-implantation embryo is required to induce the formation of pre-PGCs in the early epiblast and subsequently at later stages establishes the posterior signalling niche necessary for PGC specification.

Imaging studies demonstrate that PGCs actively migrate as individual cells from the base of the allantois towards the overlying endoderm[54]. However, the molecular pathways guiding this initial directional migration remain ill-defined. Besides intrinsic regulators of cell motility the posterior endoderm also plays a critical role guiding PGC migration. For example, in Sox17 mutant embryos that display defective endoderm formation, PGCs are formed, mature and initiate migration towards the endoderm but lack the ability to appropriately integrate to the endoderm[55]. Similarly we found here in Smad2ΔVE and Smad4ΔEpi embryos that PGCs are appropriately specified, but fail to migrate towards the endoderm. PGCs normally extend long filopodia-like structures at e7.5[54]. In contrast in Smad2ΔVE embryos endoderm formation is compromised, filopodia-like structures are never observed and PGCs remain as clusters of epithelial-like cells. Likewise, Smad4ΔEpi embryos fail to form DE[39]. Migration defects could potentially reflect the intrinsic loss of Smad4-dependent PGC functional activities, or alternatively, defective endoderm formation potentially results in the failure to produce chemo-attractants. The extracellular matrix (ECM) also plays a crucial role in guiding PGC migration[56,57]. We previously reported that Smad4 controls ECM deposition in early developmental stages[58]. Further studies on the trophic signals emitted from the posterior endoderm and the potential role of Smad4-dependent signalling in ECM composition will be required to further define Bmp/Nodal Smad requirements during PGC migration.

An antagonistic relationship between Nodal and Bmp pathways has been described in a variety of developmental contexts including within the VE for establishing initial proximal-distal polarity, patterning of the PS, morphogenesis of the amnion and correct establishment of the left–right body plan[9,10,13,59,60]. The present experiments further demonstrate that these regulatory cues govern cell fate decisions in the posterior epiblast causing a discrete subpopulation to adopt a germ cell versus somatic cell fate. Importantly, we found that Blimp1 induction within the PGCs is associated with down-regulated pSmad159 levels providing a cell intrinsic regulatory mechanism that allows this discrete subpopulation to become non-responsive to local Bmp signalling cues. Future experiments will be needed to further define dynamic cellular events controlling this developmental switch.

## Methods

**Animal care.** *Smad2⁻/⁻* [25], *Smad2^CA* [61], *Nodal⁻/⁻* [24], *Eomes⁻/⁻*, *Eomes^CA* [36], *Smad4⁻/⁻*, *Smad4^CA* [39], *Prdm1⁻/⁻* [4], *Prdm1^CA* [62], *Smad1⁻/⁻*, *Smad1^CA* [15], *Prdm1^Cre-Lacz* [40], *Ttr-Cre* [26], *Sox2-Cre* [33], *Prdm1-mVenus* [32] alleles were genotyped as described. Supplementary Table 2 indicates inter-crosses produced for this study and acronyms used for embryos. All animal experiments were performed in accordance with Home Office (UK) regulations and approved by the University of Oxford Local Ethical Committee.

The sex of the individual embryos was not determined—over the course of the experiments we assume that the ratio of males to females is 1:1 reflecting the sex ratio of the pups at birth.

**Generation of knockout ES cell lines and ES cell culture.** Blastocysts for ESC derivation were obtained from inter-crosses of *Wnt3⁺/⁻* [23], *Smad1⁺/⁻* [15], *Smad2⁺/⁻* [25], *Smad4⁺/⁻* [39] and *Eomes⁺/⁻* [36] mice harbouring the *Prdm1-mVenus* BAC transgene[32]. All ESC lines used were grown in feeder-free conditions on 0.1% gelatin-coated dishes at 6% $CO_2$ at 37 °C. ESCs were cultured in serum-free media

containing N2B27 (Takara, Y40002) supplemented with 1 μM PD0325091 and 3 μM CHIR99021 and 1000 U/ml LIF.

**PGCLC cultures.** EpiLCs and PGCLCs were induced as previously described[63]. In brief, $7 \times 10^5$ ESCs were washed and resuspended in N2B27 medium (Takara, Cat#Y40002) supplemented with 12 ng/ml bFgf (Invitrogen, 13256-029), 20 ng/ml Activin A and 1% KSR (Gibco, 10828, Lot:1508151) and grown on fibronectin-coated (10 μg/cm²) (Millipore, FC010) 6 cm dishes for 48 h to form EpiLCs. 2000 EpiLCs were then washed and plated into ultra-low attachment U-bottom shaped 96-well plates in serum-free medium (GK15; GMEM (Invitrogen) with 15% KSR, 0.1 mM NEAA, 1 mM sodium pyruvate, 0.1 mM 2-mercaptoethanol, 100 U/ml penicillin, 0.1 mg/ml streptomycin, and 2 mM L-glutamine) in the presence of the cytokines BMP4 (500 ng/ml; R&D Systems, 314-BP), LIF (1000 U/ml; Millipore, ESG1107), SCF (100 ng/ml; R&D Systems, 455-MC), BMP8b (500 ng/ml; R&D Systems, 1073-BP) and EGF (50 ng/ml; R&D Systems, 2028-EG). Differentiations were performed in two independent clonal cell lines per genotype.

**Immunofluorescence.** d2 and d4 PGCLCs as well as E5.5–E8.5 mouse embryos were harvested, washed in phosphate-buffered saline (PBS) and fixed in 1% par-aformaldehyde (PFA) o/n at 4 °C. After three washes in PBS containing 0.1% Triton X-100 (PBS-Tx), samples were permeabilised in PBS containing 0.5% Triton X-100 followed by three washes in PBS-Tx, blocked in 5% donkey serum and 0.2% BSA in PBS-Tx for 1 h at RT and incubated overnight with primary antibodies in blocking solution at 4 °C. Following four washes in PBS-Tx, samples were incubated with fluorophore-conjugated secondary antibodies in blocking solution (2 h, RT) followed by three-five washes in PBS-Tx, one wash in PBS-Tx containing 2 μg/ml DAPI and three washes in PBS-Tx prior to mounting in Vectashield with DAPI (H-1200). Samples were imaged the following day on an Olympus Fluoview FV1000 confocal microscope and image data was processed using ImageJ and Bitplane Imaris software. Antibodies are listed in the Supplementary Table 3. Multiple litters of each intercross were examined, with a minimum of $n = 3$ mutant embryos stained and imaged.

**RT-qPCR.** Two hundred and fifty nanogram RNA was reverse transcribed to cDNA using Superscript III First Strand Synthesis System (Life Technologies, Cat#18080-051) and diluted to 100 μl final volume in H₂O (2.5 ng/ul). Two microlitre (5 ng) cDNA were used per RT-qPCR reaction in duplicate using SYBR-green kit (Qiagen, Cat#204143). Relative gene expression was normalised to Gapdh expression and calculated as $2^{\Delta\Delta Ct}$. Average expression levels were calculated for two technical replicates from two independent cell lines per genotype. RT-qPCR primer sequences are listed in Supplementary Table 4.

**In situ hybridisation and immunohistochemistry.** Whole-mount in situ hybridisation analysis was performed as before[50], using antisense riboprobes for Bmp2[64], Bmp4[13] and Gdf3[28]. For Blimp1 immunohistochemistry, E7.5 decidua were fixed overnight in 4% PFA, dehydrated in ethanol, embedded in paraffin wax and sectioned (6 μm). Dewaxed sections were subjected to antigen retrieval by boiling for 1 h in Tris/EDTA (pH 9.0) and permeabilized for 10 min in 0.1% Triton X-100 in TBS. Sections were subsequently blocked with 10% normal goat serum in TBS. Sections were incubated with rat monoclonal anti-Blimp1 (1:500 dilution, sc-130917, Santa Cruz Biotechnology) in block overnight at 4 °C and signal-amplified with rabbit anti-rat secondary antibody (AI-4001, Vector Laboratories) for 45 min at RT followed by peroxidase blocking for 20 min at RT and development with Envison System-HRP for rabbit antibodies (K4011, DAKO) and Vector Red substrate (SK-4805, Vector Laboratories). Sections were lightly counter stained with haematoxylin, coverslipped and imaged.

Haematoxylin and eosin staining was performed as previously described[36].

**Reporting summary.** Further information on experimental design is available in the Nature Research Reporting Summary linked to this article.

## Data availability

The authors declare that all data supporting the findings of this study are available within the article and its supplementary information files or from the corresponding author upon reasonable request. Publicly available source data in Fig. 5 and Supplementary Figure 5 were obtained from GSE91041, GSE91040 and E-MTAB-1178.

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

## Acknowledgements

We would like to thank Mitinouri Saitou for generously providing the Blimp1-mVenus BAC transgenic mouse strain. Confocal microscopy was carried out in the Micron Advanced Bioimaging Unit (funded by Wellcome Trust Strategic Award 107457). This work was funded by the Wellcome Trust (099840/Z/12/A to A.D.S. and 102811/Z/13/2 to E.J.R.). E.J.R. is a Wellcome Trust Principal Research Fellow.

## Author contributions

All the authors designed the experiments, A.D.S. and I.C. performed the experiments. A.D.S., I.C., E.K.B and E.J.R contributed to writing the paper.

## Additional information

**Competing interests:** The authors declare no competing interests.

