## [Peer Review File · Nature Communications]

Reviewers' Comments:

Reviewer #1:

Remarks to the Author:

The authors have investigated the precise role of Nodal and BMP signaling in primordial germ cell (PGC) development, by conditional deletion of several genes (Smad2, Nodal, Eomes, Smad4, Smad1) in the epiblast (EPI) or visceral endoderm (VE). Extensive genetic dissection study has now revealed exactly where, when and how each signaling is required for PGC specification, maintenance or migration. Thus, BMP-Smad1 signaling in EPI is essential for induction of pre-PGCs positive for Blimp1. Nodal-Smad2 signaling in VE restricts BMP signaling to the proximal region of the embryo via the antagonistic relationship between Nodal and BMP pathways, thereby determines the position of PGC at the proximal region. Nodal-Eomes pathway in EPI is dispensable for pre-PGC induction but is required to form specified PGCs positive for PGC markers such as Stella. Interestingly, Blimp1-Venus+ cells (pre-PGCs~PGCs) are negative for pSmad159. The level of pSmad159 and Smad1 expression in pre-PGC ~PGC is repressed by Blimp1, since 1) pSmad159 was up-regulated in Blimp1-Venus+ cells of Blimp1(-/-) embryos, 2) Smad1 expression is down-regulated during PGC development, but failed to be repressed in the absence of Blimp1, and 3) Blimp1 ChIP-Seq analysis found a Blimp1-binding peak and a binding motif in the 1st intron of Smad1 gene. Thus, Blimp1-regulated BMP signaling, which in turn antagonizes Nodal signaling, influences the cell fate decision between germ cells and somatic cells. In all, the paper provides a previously unknown mechanism that would address how the germ cell vs. somatic cell decision is made during development. The paper can be published after minor revision.

Minor comments

1. In Figure 5D, cells indicated by the arrows seem negative for pSmad159. It is more informative to mark BV+pSmad159+ cells of the Blimp null embryo (such as those found in the upper part of the panel), since this is the point of this figure.
2. Presumably, Smad4 mediates both Smad1 and Smad2 signaling. Phenotype of Smad4 Δ EPI may be discussed in comparison to that of Smad2 Δ EPI and Smad1 Δ EPI.
3. Blimp1-Venus (more precisely, Blimp1-mVenus) transgene encodes a membrane-localized Venus protein, which is the reason why the cell membrane is stained in this paper. This needs to be mentioned in Results (page 6).

Reviewer #2:

Remarks to the Author:

This manuscript explores the regulatory signals controlling cell fate decision during early PGC specification and development. This is done using Conditional knock-out mouse models. This shows that loss of Smad2 in the VE phenocopies a constitutive Smad2 knock-out. The authors show that Nodal/Smad2 confines BMP signaling from EXE to spatially restrict the pre-PGC niche. Knockout of Nodal or Smad2 extends the region of active BMP signaling (marked by pSmad159) distally. Moreover, both knockouts show earlier PGC initiation with more pre-PGC founders present in the Smad2DVE KO. For Smad2DVE, this is not due to increased levels of BMP4/2 mRNA expression, suggesting antagonism between Nodal and BMP may be involved. Indeed, Smad2DVE have decrease levels of the BMP antagonist GDF3 in both VE and epiblast. Smad2DVE also mispatterns Nanog/Oct4; Blimp1-venus, with Blimp1+ cells appearing 12-18 hours earlier than in wild-type and at E6.5 being present at higher numbers. Smad2DVE have increased numbers of Blimp-Venus, Oct4, Nanog Stella cells. Therefore Smad2 in the VE restricts the PGC number. The authors show that Eomes is important for PGC specification, possibly because of a role in formation of the posterior signaling niche. The authors also present data suggesting that Smad1 is important for PGC specification, while co-effector Smad4 for maintenance and specification of PGCs. Interestingly, the authors find that after PGCs specification, the activity of pSmad159 decreases to ensure PGCs become refractory to BMP signaling. This work therefore includes novel findings that

will influence thinking in the field and that will be of general interest.

However, there are several issues that should be addressed to improve the likely impact of the work.

- 1) A general point is that I feel the results section is more difficult to read than it should be. This may at least partly be because the paper covers so much ground. A little more signposting of the logic to introduce individual experiments in the results section would help. At times it seemed like experiments are being presented haphazardly. The tendency to intersperse positive results with negative data (which are given undue prominence due to injudicious placement within the text) also detracted from the main findings.
- 2) Although, Nodal^{-/-} ESCs were reported have a PGCLC induction defect in vitro, both Smad2^{-/-} ΔVE and ΔEpi can initiate PGC specification. As Nodal^{-/-} ΔEpi have BV⁺ cells, the authors should analyse these BV⁺ cells at a later stage to check whether they express, for example, Oct4, Stella normally. It is also important to KO Smad2 or Nodal in germ cells (Blimp1Cre) to confirm that the germ cell development defect is not because of a missing intrinsic Nodal signal.
- 3) There is no analysis of Stella expression to support the idea that mature Stella positive PGCs fail to emerge in Smad1^{-/-} embryos. Smad1^{-/-} ΔEpi seems have small numbers of Oct4⁺ PGCs at E8.5 (Fig 3C). Smad1^{-/-} ESCs can generate PGCLCs or initiate PGC cells to a lower extent. This might be because of compensation by Smad5; the authors should discuss this possibility.
- 4) on page 9, the authors conclude paragraph 1 saying 'Smad4 is dispensible for initial specification but is required for maintenance and/or migration'. However, from the images in Fig 3A, it seems just as likely that there is a delay in development of Smad4ΔEpi embryos. Can the authors refute this possibility? Also, the in vitro PGCLCs experiment shows that Smad4^{-/-} ESCs yield fewer BV⁺ and Oct4⁺ cells and mRNA expression as early as day 2. How do the authors explain this?
- 5) The authors raise an interesting point that Eomes might be not be intrinsically required for PGC formation but rather, be important for the formation of posterior signaling niche. Blimp1KO Eomes should be performed to confirm this.
- 6) Meta-analysis of RNA data shows that Blimp1 stimulates a modest downregulation of Smad1 and Smad5 after PGC specification. During PGCLC differentiation, BLIMP1 comes on as early as day 2. However, the authors only show negative signal of pSmad159 in BV⁺ cells at day 6 of PGCLC differentiation; what about at day 2? The authors should perform early Smad1, 5, 9 mRNA or pre-mRNA changes by ectopically inducing Blimp1 expression during PGCLCs to provide more convincing evidence that Blimp1 directly represses Smad1.
- 7) on page 7 (last paragraph) the text is somewhat confusing as the Blimp-Venus and Nanog results are inaccurately simplified. Fig2a shows there are 2 populations of Blimp-venus expressing cells. These include BV⁺⁺, that are Nanog⁻, but also BV⁺ cells that are proximal and are Nanog positive. This section should be re-written for clarity.
- 8) the idea that BV high cells partially retain pSmad159 reactivity is weak and is used to justify the data in fig 5E, which leads to the authors over-claiming that Blimp1 downregulates Smad1 expression at early stages of PGC development etc.' This statement requires the authors to remove the Blimp1 motif from the Smad1 gene and demonstrate retained BMP-responsiveness. If this is not possible, the authors should rephrase the final sentence of the results to emphasise its speculative nature.

Minor issues,

- 1) no scale bars, no replicate numbers
- 2) more precise mouse embryos staging, preferably using Downs and Davies (eg, Fig 1a and Fig 2a, both are called E5.5. However, the p-Smad159 is different).
- 3) zoom in of some figures eg, Fig S1C, FigS2E, Fig2D, Fig5A.
- 4) page 4, last sentence, there is no Nanog co staining data
- 5) page 10 "Similarly, Eomes..." Eomes ΔEpi PGC should arguably not be mentioned here, as they are not normal PGCs.
- 6) Discussion part, page10 "expanded Nodal activity...", expanded?
- 7) in 'conditional inactivation of...', I would suggest moving the last 2 paragraphs up, so that the section can end on a strong note (Smad2 in the VE can restrict expansion of the PGC niche).

- 8) Supp fig 1E and Supp fig 2E are both labelled as e6.5 and this should be corrected.
- 9) on page 8 the final 3-line paragraph should be moved up, to end the section on a positive note.
- 10) I think the section 'Smad1 is required for specification...' would flow better in the narrative if the Smad4 and Smad1 sub-sections were inverted.
- 12) on page 10, the authors refer to RT-qPCR. If this is Fig 4C, the legend should say that this is RT-qPCR.
- 13) the sentence including 'exception of Smad2^{-/-} and Eomes^{-/-} cultures' should be amended to also include Smad4^{-/-} or else the wording should be changed.
- 14) The sentence including 'outside the context of the early embryo' might be easier to read if changed to 'in contrast to the situation in vivo'
- 15) the authors claim that Fig5A shows that 'BV+ cells themselves are devoid of p-Smad159 staining'; not at the magnification shown it doesn't.
- 16) the authors state that 'specified PGCs do not respond to high levels of BMP signalling'. How does this relate to the period of competence?
- 17) when decreased p-Smad159 is seen, what is happening to the surface expression of BMP receptors?

Point-by-Point Response to the Reviewers' comments

Reviewer #1 (Remarks to the Author):

The authors have investigated the precise role of Nodal and BMP signaling in primordial germ cell (PGC) development, by conditional deletion of several genes (Smad2, Nodal, Eomes, Smad4, Smad1) in the epiblast (EPI) or visceral endoderm (VE). Extensive genetic dissection study has now revealed exactly where, when and how each signaling is required for PGC specification, maintenance or migration. Thus, BMP-Smad1 signaling in EPI is essential for induction of pre-PGCs positive for Blimp1. Nodal-Smad2 signaling in VE restricts BMP signaling to the proximal region of the embryo via the antagonistic relationship between Nodal and BMP pathways, thereby determines the position of PGC at the proximal region. Nodal-Eomes pathway in EPI is dispensable for pre-PGC induction but is required to form specified PGCs positive for PGC markers such as Stella. Interestingly, Blimp1-Venus+ cells (pre-PGCs~PGCs) are negative for pSmad159. The level of pSmad159 and Smad1 expression in pre-PGC~PGC is repressed by Blimp1, since 1) pSmad159 was up-regulated in Blimp1-Venus+ cells of Blimp1(-/-) embryos, 2) Smad1 expression is down-regulated during PGC development, but failed to be repressed in the absence of Blimp1, and 3) Blimp1 ChIP-Seq analysis found a Blimp1-binding peak and a binding motif in the 1st intron of Smad1 gene. Thus, Blimp1-regulated BMP signaling, which in turn antagonizes Nodal signaling, influences the cell fate decision between germ cells and somatic cells. In all, the paper provides a previously unknown mechanism that would address how the germ cell vs. somatic cell decision is made during development. The paper can be published after minor revision.

Minor comments

1. In Figure 5D, cells indicated by the arrows seem negative for pSmad159. It is more informative to mark BV+pSmad159+ cells of the Blimp null embryo (such as those found in the upper part of the panel), since this is the point of this figure.

We thank the reviewer for this suggestion, and have re-annotated the Panel as requested. We have included arrowheads to indicate BV+ cells that retain p-Smad159 reactivity, and reworded the relevant sentence in the Results section p 11 to clarify this point. It now reads "Interestingly a subset of BV+ cells display decreased p-Smad159 levels, while others BV+ cells retain p-Smad159 reactivity (Fig. 5D)" We have also amended Figure 5 legend to read: "(D) p-Smad159 staining in control and Blimp1-/- BV+ embryos at e7.5. Arrows indicate reduction of p-Smad159 reactivity in BV-high expressing cells in expanded panels. Arrowheads indicate retained p-Smad159 activity in BV+ cells in the Blimp1-/- mutant embryo."

2. Presumably, Smad4 mediates both Smad1 and Smad2 signaling. Phenotype of Smad4 Δ EPI may be discussed in comparison to that of Smad2 Δ EPI and Smad1 Δ EPI.

Yes, this is an interesting point. Considerable evidence suggests both Smad2/3 and Smad1/5 activities are Smad4 independent in a variety of tissue contexts. For example we previously described that Smad4 epiblast deletion (Chu et al 2004) has no effect on gastrulation. Similarly here if Smad4 was a critical partner of Smad1, Smad4 epiblast deletion would prevent PGC formation. Rather PGCs specification is Smad4 independent but they fail to be maintained/migrate.

To this point we have now briefly discussed the Smad4 Δ Epi phenotype in relation to Smad2 Δ Epi and Smad1 phenotypes in the Results section p 9. We also conclude that "Hence the Smad4 Δ Epi mutant phenotype is intermediate to the Smad2 Δ Epi and Smad1^{-/-} phenotypes, as in Smad2 Δ Epi embryos Stella⁺ cells form in the posterior side, while Smad1 mutants initiate pre-PGC differentiation but never develop specified PGCs."

3. Blimp1-Venus (more precisely, Blimp1-mVenus) transgene encodes a membrane-localized Venus protein, which is the reason why the cell membrane is stained in this paper. This needs to be mentioned in Results (page 6).

We have inserted “membrane tethered” in the Results (p 6) where the Blimp1-mVenus (BV) transgene is introduced and modified Supplemental Table S2 accordingly.

--

Reviewer #2 (Remarks to the Author):

This manuscript explores the regulatory signals controlling cell fate decision during early PGC specification and development. This is done using Conditional knock-out mouse models. This shows that loss of Smad2 in the VE phenocopies a constitutive Smad2 knock-out. The authors show that Nodal/Smad2 confines BMP signaling from EXE to spatially restrict the pre-PGC niche. Knockout of Nodal or Smad2 extends the region of active BMP signaling (marked by pSmad159) distally. Moreover, both knockouts show earlier PGC initiation with more pre-PGC founders present in the Smad2DVE KO. For Smad2DVE, this is not due to increased levels of BMP4/2 mRNA expression, suggesting antagonism between Nodal and BMP may be involved. Indeed, Smad2DVE have decrease levels of the BMP antagonist GDF3 in both VE and epiblast. Smad2DVE also mispatterns Nanog/Oct4; Blimp1-venus, with Blimp1+ cells appearing 12-18 hours earlier than in wild-type and at E6.5 being present at higher numbers. Smad2DVE have increased numbers of Blimp-Venus, Oct4, Nanog Stella cells. Therefore Smad2 in the VE restricts the PGC number. The authors show that Eomes is important for PGC specification, possibly because of a role in formation of the posterior signaling niche. The authors also present data suggesting that Smad1 is important for PGC specification, while co-effector Smad4 for maintenance and specification of PGCs. Interestingly, the authors find that after PGCs specification, the activity of pSmad159 decreases to ensure PGCs become refractory to BMP signaling. This work therefore includes novel findings that will influence thinking in the field and that will be of general interest.

However, there are several issues that should be addressed to improve the likely impact of the work.

1) A general point is that I feel the results section is more difficult to read than it should be. This may at least partly be because the paper covers so much ground. A little more signposting of the logic to introduce individual experiments in the results section would help. At times it seemed like experiments are being presented haphazardly. The tendency to intersperse positive results with negative data (which are given undue prominence due to injudicious placement within the text) also detracted from the main findings.

During our final editing of the text to comply with the word limit we removed numerous transition sentences. At the reviewers suggestion, and with the editors discretion, we have now added back a few more sentences throughout the Results (p 6, p 7, p 8) to further explain our rationale behind the experimental design.

2) Although, Nodal^{-/-} ESCs were reported have a PGCLC induction defect in vitro, both Smad2^{-/-} ΔVE and ΔEpi can initiate PGC specification. As Nodal^{-/-} ΔEpi have BV⁺ cells, the authors should analyse these BV⁺ cells at a later stage to check whether they express, for example, Oct4, Stella normally. It is also important to KO Smad2 or Nodal in germ cells (Blimp1Cre) to confirm that the germ cell development defect is not because of a missing intrinsic Nodal signal.

We have now analysed BV/Nanog at e6.5 and BV/AP2 γ staining in e6.5 and e7.5 Nodal null embryos and have included this data in Supplementary Figure 2 Panel E, F and G. These new data have been presented in the final 4 sentences of the Results on p7/8.

Based on these collective data we conclude on p 8 “Hence specified PGCs are not observed in Nodal $^{-/-}$ embryos”.

We also examined Stella expression at e6.5 and, as for Ap2 γ , we did not observe premature Stella expression in BV+ cells. We include this data for the reviewer below.

Reviewer Figure 1: Stella staining in e6.5 control and Nodal null BV+ embryos.

We respectively disagree that deleting either Smad2 or Nodal from the PGCs would be informative. Clearly Smad2 is not required since an epiblast completely devoid of Smad2 forms numerous mature Stella+ PGCs (Supp Fig.S2A). Smad2 and Smad3 are functionally redundant and co-expressed in the epiblast (Tremblay et al., 2000; Dunn et al., 2004) and we have no ready means of removing both Smad2 and Smad3 from the PGCs. Removing Nodal from the PGCs will be inconclusive because the surrounding wild-type epiblast cells make abundant levels of Nodal ligand. Importantly we previously showed Nodal deficient ES cells contribute normally in chimeric embryos because they are rescued by extrinsic signaling provided by adjacent wild type cells (Conlon et al., 1991; Varlet et al., 1997).

3) There is no analysis of Stella expression to support the idea that mature Stella positive PGCs fail to emerge in Smad1 $^{-/-}$ embryos. Smad1 $^{-/-}$ Δ Epi seems have small numbers of Oct4+ PGCs at E8.5 (Fig 3C). Smad1 $^{-/-}$ ESCs can generate PGCLCs or initiate PGC cells to a lower extent. This might be because of compensation by Smad5; the authors should discuss this possibility.

We agree completely about Smad5 compensation and have addressed this on p 9 in the revised manuscript. “The ability to generate some BV/Oct4⁺ cells may be due to Smad5 compensation, which is also expressed in the epiblast of gastrulating embryos^{15,20}. Indeed, in addition to reduced PGC numbers in embryos lacking either Smad1 or Smad5, Smad1/5 double heterozygotes also have a reduced number of PGCs²⁰, revealing a critical role for both Smad1 and Smad5 in germ cell development”.

We have attempted to stain numerous e8.5 Smad1 mutant embryos for Stella, but cannot detect any Stella positive cells at this stage of development – we elected not to include this negative data.

4) on page 9, the authors conclude paragraph 1 saying ‘Smad4 is dispensible for initial specification but is required for maintenance and/or migration’. However, from the images in Fig 3A, it seems just as likely that there is a delay in development of Smad4DEpi embryos. Can the authors refute this possibility? Also, the *in vitro* PGCLCs experiment shows that Smad4^{-/-} ESCs yield fewer BV⁺ and Oct4⁺ cells and mRNA expression as early as day 2. How do the authors explain this?

Comparable numbers of BV+Stella+ cells are present at e7.5 in both control and Smad4ΔEpi embryos (Suppl Fig S3B), showing that PGC specification in the epiblast is Smad4-independent. However, by e8.5 these specified PGCs have not expanded or migrated to the posterior endoderm. Since Stella+ cells are evident at e7.5 we do not believe there is a delay in PGC specification, rather we conclude that the role of Smad4 lies in the following 24 hours when PGCs are required to further expand and migrate. For clarity the Smad4ΔEpi embryos are not developmentally delayed compared to their littermates.

The reduced number of BV+ cells formed *in vitro* in PGCLC cultures likely results from a secondary defect, as Smad4 is also required for visceral endoderm development (Sirard et al 1998). We previously showed that growth and development of Smad4^{-/-} EBs is compromised due to inappropriate endoderm development (Costello et al 2009), which could secondarily affect BV+ cell numbers formed in the 3D aggregates. The protocol generates a mix of primordial germ cell-like cells and other supporting cell populations, so we only conclude that specified PGCLC (Stella+BV+ or Ap2γ+BV+) can be found at stages comparable to the wildtype control cultures. It’s not possible to assess kinetics of PGCLC differentiation in Smad4 deficient cultures due to Smad4 requirements in additional cell types. However, our conclusions from the PGCLC experiments support our *in vivo* findings that Smad4 is not required for formation of specified PGCs (BV+Stella+ or BV+Ap2γ+).

5) The authors raise an interesting point that Eomes might be not be intrinsically required for PGC formation but rather, be important for the formation of posterior signaling niche. Blimp1KO Eomes should be performed to confirm this.

We agree this is a potentially interesting question but unfortunately the proposed experiment is flawed because Eomes is co-expressed with the Blimp1-Cre deleter in the visceral endoderm (Vincent et al, 2005; Nowotschin et al, 2013), and removing Eomes from the VE severely disturbs the ability of the E5.5 embryo to establish early A-P pattern (Nowotschin et al 2013), potentially leading to indirect effects on PCG formation. Other PGC-Cre deleter strains (e.g. Stella.Cre) are only active following PGC specification and would not address Eomes early requirements.

6) Meta-analysis of RNA data shows that Blimp1 stimulates a modest downregulation of Smad1 and Smad5 after PGC specification. During PGCLC differentiation, BLIMP1 comes on as early as day 2. However, the authors only show negative signal of pSmad159 in BV+

cells at day 6 of PGCLC differentiation; what about at day 2? The authors should preform early Smad1, 5, 9 mRNA or pre-mRNA changes by ectopically inducing Blimp1 expression during PGCLCs to provide more convincing evidence that Blimp1 directly represses Smad1.

As requested we now provide further data showing staining of p-Smad159 in BV+ cells in both d2 and d4 PGCLC cultures (Suppl Fig. 5B). Interestingly as early as d2 BV+ cells have little to no p-Smad159 staining, further supporting our hypothesis that BV+ cells down-regulate BMP signalling, compared to the surrounding BV- cells within the culture.

The reviewer also suggests we should try over-expressing Blimp1 in PGCLC cultures. Unfortunately multiple labs (including our own) have attempted ectopic Blimp1 expression in a variety of cell types. There is widespread agreement that excessive Blimp1 expression is highly toxic. The Surani lab transiently transfected P19 cells and FACS sorted weak expressors for ChIP-seq experiments (Magnusdottir et al, 2013). PGCLC cultures are generated by seeding 2000 cells/well in 96 well trays in media containing a cocktail of growth factors, to form very focal cell clumps. Also, only ~20-30% of cells within the 3D aggregates become BV+ PGCLCs, with the remainder apparently forming a “PGC supporting cell niche” that retain Bmp-signalling. Devising protocols to induce Blimp1 in a controlled fashion in these cultures would be technically challenging.

7) on page 7 (last paragraph) the text is somewhat confusing as the Blimp-Venus and Nanog results are inaccurately simplified. Fig2a shows there are 2 populations of Blimp-venus expressing cells. These include BV++, that are Nanog -, but also BV+ cells that are proximal and are Nanog positive. This section should be re-written for clarity.

Apologies for the confusion, we have now re-written this section to better clarify the result we were attempting highlight namely that there is either no or only weak re-expression of Nanog in the BV+ cells, regardless of BV expression levels, as BV and Nanog are expressed in pre-PGCs prior to PGC specification. Accordingly we have modified this paragraph as follows:

“These BV⁺ cells have either weak or no Nanog expression at e5.5 (Fig. 2A). By e6.5 there is no expression of Nanog in the BV⁺ epiblast cells or indeed any other cells of the epiblast (Supplementary Fig. 2E).”

8) the idea that BV high cells partially retain pSmad159 reactivity is weak and is used to justify the data in fig 5E, which leads to the authors over-claiming that Blimp1 downregulates Smad1 expression at early stages of PGC development etc.’ This statement requires the authors to remove the Blimp1 motif from the Smad1 gene and demonstrate retained BMP-responsiveness. If this is not possible, the authors should rephrase the final sentence of the results to emphasise its speculative nature.

Retained p-Smad159 activity in BV+ cells in the Blimp1 mutants (Figure 5D) was not solely used to suggest this as a possible contributory mechanism. We show that not all BV+ cells have reduced levels of p-Smad159. Some have reduced levels (arrows), but not to the same extent as wildtype, while other BV+ cells still have clear p-Smad159 reactivity (now shown as arrowheads in Figure 5D).

Given reduced p-Smad159 levels in specified wild type PGCs, we wanted to determine if this reflected decreased transcriptional output. Previously published datasets, shown in Supplementary Figure 5 C & D show lower levels of *Smad1* in the PGCs as compared to the surrounding somatic cells (Supplementary Figure 5C). Moreover as PGCs become specified and migrate, there is a further reduction in *Smad1* transcripts compared to the surrounding somatic cells (Supplementary Figure 5D). This reduction in *Smad1* expression levels during PGC specification does not occur in the compromised PGC cells formed in Blimp1 mutants (Supplementary Figure 5D). Collectively these published observations led

us to speculate that Blimp1 may be directly repressing *Smad1* expression as we observe a ChIP-Seq peak containing a perfect Blimp1 consensus binding motif in d6 PGCLCs.

We also provide below additional data for the reviewer to show this peak is also observed in other Blimp1-dependent tissues, to strengthen our hypothesis that Blimp1 binding may contribute to repressing *Smad1* expression. However the suggested experiment, namely to delete the peak from the *Smad1* locus, is technically complicated and will be very time consuming.

Reviewer Figure 2: Genome browser track view of GFP-ChIP wiggle plots from GFP-Blimp1 (Mitani 2017) or Blimp1-GFP (Mould 2015) expressing cells of indicated tissues showing enrichment of GFP-tagged Blimp1 at a *Smad1* intronic regions. The sequence below the ChIP peak contains a Blimp1 motif indicated in red.

As suggested by the reviewer we have modified the relevant parts of the text to reflect that this is merely an attractive hypothesis. We have been careful to modify the language in the Results section p 11 to state “Interestingly a sub-set of BV+ cells display decreased p-Smad159 levels while others retain p-Smad159 reactivity (Fig 5D)”. Our final sentence in this section has been edited to state “Thus, Blimp1 may downregulate *Smad1* expression at early stages of PGC development, potentially ensuring that the emerging PGCs are refractory to the high Bmp signalling environment in the posterior epiblast that normally promotes development of the ExM.”

Minor issues,

1) no scale bars, no replicate numbers

These have been added as requested

2) more precise mouse embryos staging, preferably using Downs and Davies (eg, Fig 1a and Fig 2a, both are called E5.5. However, the p-Smad159 is different).

Downs & Davies (1993) is a very rough staging system and only includes one “pre-gastrulation” stage embryo (single image and one hand drawn cartoon). We agree that since the embryo is growing rapidly it’s difficult to precisely “stage”, but there are no tangible morphological features that allow embryos to be more accurately staged. If using the Downs and Davies staging, the images in Fig1a and Fig2b would both still be referred as the same label i.e. pre-streak (PS), so it would not affect the sub-classification of pre-streak embryos. To simplify the data presented, and to make it more accessible to a wider readership, we used the timing of dissections relative to the mid-point of the dark-cycle. In all our experiments littermates were used as controls.

3) zoom in of some figures eg, Fig S1C, FigS2E, Fig2D, Fig5A.

As requested all of these images have now been included in the revised figures.

4) page 4, last sentence, there is no Nanog co staining data

We show Nanog and Oct4 co-staining in Figure 1C in the centrally located PGC-like cells at e6.5 in *Smad2* Δ VE mutants. This is why we used the wording “Overall...”, as we are considering data from both e6.5 and e7.5 *Smad2* Δ VE embryos. Stella co-staining is only shown at e7.5, while Brachyury is shown at e6.5.

5) page 10 “Similarly, Eomes...” Eomes Δ Epi PGC should arguably not be mentioned here, as they are not normal PGCs.

We have edited this to state “Additionally, Eomes Δ Epi embryos also display a high proportion of BV+ epiblast cells lacking p-Smad159 activity”.

6) Discussion part, page10 “expanded Nodal activity...”, expanded?

Apologies for lack of explanation. In *Smad2*^{-/-} embryos there is an expansion of Nodal activity, as judged by Nodal expression throughout the epiblast (Waldrip 1998), rather than restricted to the posterior side of the embryos, as occurs in wildtype embryos.

We have edited this section (p 12) to read “Interestingly these disturbances in *Smad2* Δ VE mutants are associated with decreased expression of the Bmp antagonist *Gdf3* and likely accounts for the increased levels of p-Smad159 in the VE”.

7) in ‘conditional inactivation of...’, I would suggest moving the last 2 paragraphs up, so that the section can end on a strong note (*Smad2* in the VE can restrict expansion of the PGC niche).

While we appreciate this suggestion, we prefer to retain the current structure as we first discuss the *Smad2* Δ VE mutant and the expansion of the Bmp signalling domain.

8) Supp fig 1E and Supp fig 2E are both labelled as e6.5 and this should be corrected.

Supplemental Figure S2I has now been altered to e7.0, as this is a mid to late streak embryo. Supplemental Figure S1E is labelled e6.5 as this is an early to mid-streak embryo.

9) on page 8 the final 3-line paragraph should be moved up, to end the section on a positive note.

We have now moved this paragraph to the start of the Eomes section as suggested.

10) I think the section ‘*Smad1* is required for specification...’ would flow better in the narrative if the *Smad4* and *Smad1* sub-sections were inverted.

Again, while we appreciate this suggestion, we first discuss Smad4 as it's involved in both Nodal and Bmp signalling, and forms a logical link between the Nodal signalling (presented first) and the subsequent Bmp/Smad1 data. We prefer to retain the original order.

11)

There was no 11th comment

12) on page 10, the authors refer to RT-qPCR. If this is Fig 4C, the legend should say that this is RT-qPCR.

The legend has been corrected.

13) the sentence including 'exception of Smad2^{-/-} and Eomes^{-/-} cultures' should be amended to also include Smad4^{-/-} or else the wording should be changed.

We thank the reviewer for pointing this out. For clarification we have edited this part of the results on p 10 to read "Thus, Smad2^{-/-}, Eomes^{-/-} and Smad4^{-/-} cell aggregates express lower levels of *Prdm1* (Blimp1) and *Tfap2c* (Ap2γ), while Smad1^{-/-} cultures show reduced *Tfap2c* and increased levels of the somatic marker *Hoxb1* (Fig. 4C)"

14) The sentence including 'outside the context of the early embryo' might be easier to read if changed to 'in contrast to the situation in vivo'

We have modified the sentence as requested.

15) the authors claim that Fig5A shows that 'BV+ cells themselves are devoid of p-Smad159 staining'; not at the magnification shown it doesn't.

We have now included a panel of larger magnifications and modified the sentence in the Result to read "However, the BV+ cells themselves in e6.5 Smad2ΔVE embryos have substantially reduced levels of p-Smad159."

16) the authors state that 'specified PGCs do not respond to high levels of BMP signalling'. How does this relate to the period of competence?

Regardless of the Bmp signalling level, specified PGCs need to fully activate the germ cell transcriptional program and down-regulate the somatic program. In Blimp1 mutants few if any alkaline-phosphatase positive migrating PGCs are formed (Vincent et al, 2004). Whether this reflects the varying levels of p-Smad159 in the BV+ cells at e7.5 is unknown. All we can conclude from our data is that specified PGCs have lower levels, or are devoid of p-Smad159 staining. RNA-Seq analysis from the Mitani paper reveals that transcript levels of both Smad1 and Smad5 decrease as the PGCs develop further (e7.5 to e11.5). Hence, once pre-PGCs are formed they appear to need little to no intrinsic Bmp signalling.

17) when decreased p-Smad159 is seen, what is happening to the surface expression of BMP receptors?

This is an interesting question but not feasible with existing antibody reagents, which fail to detect endogenous membrane receptor complexes even in cultured BMP responsive cells. It's also not clear which of the numerous possible combinations of Bmp type I and type II receptors complex to transduce Bmp signalling during development of the PGCs.

Reviewers' Comments:

Reviewer #1:

Remarks to the Author:

The authors have responded to my comments properly, and have improved the manuscript. As far as this reviewer is concerned, this paper is now acceptable.

Reviewer #2:

Remarks to the Author:

The authors have answered all my concerns.

Point-by-Point Rebuttal

Since neither of the Reviewers required any further changes, we have no comments to address.

REVIEWERS' COMMENTS:

Reviewer #1 (Remarks to the Author):

The authors have responded to my comments properly, and have improved the manuscript. As far as this reviewer is concerned, this paper is now acceptable.

--

Reviewer #2 (Remarks to the Author):

The authors have answered all my concerns.